# A Multi-Agent Motion Prediction and Tracking Method Based on Non-Cooperative Equilibrium

**Yan Li, Mengyu Zhao, Huazhi Zhang, Yuanyuan Qu** and **Suyu Wang** *

School of Mechanical Electronic & Information Engineering, China University of Mining and Technology-Beijing, Beijing 100083, China; 201572@cumtb.edu.cn (Y.L.); zmy@student.cumtb.edu.cn (M.Z.); zhz@student.cumtb.edu.cn (H.Z.); 201419@cumtb.edu.cn (Y.Q.)
* Correspondence: wsy@cumtb.edu.cn; Tel.: +86-010-62331083

**Abstract:** A Multi-Agent Motion Prediction and Tracking method based on non-cooperative equilibrium (MPT-NCE) is proposed according to the fact that some multi-agent intelligent evolution methods, like the MADDPG, lack adaptability facing unfamiliar environments, and are unable to achieve multi-agent motion prediction and tracking, although they own advantages in multi-agent intelligence. Featured by a performance discrimination module using the time difference function together with a random mutation module applying predictive learning, the MPT-NCE is capable of improving the prediction and tracking ability of the agents in the intelligent game confrontation. Two groups of multi-agent prediction and tracking experiments are conducted and the results show that compared with the MADDPG method, in the aspect of prediction ability, the MPT-NCE achieves a prediction rate at more than 90%, which is 23.52% higher and increases the whole evolution efficiency by 16.89%; in the aspect of tracking ability, the MPT-NCE promotes the convergent speed by 11.76% while facilitating the target tracking by 25.85%. The proposed MPT-NCE method shows impressive environmental adaptability and prediction and tracking ability.

**Keywords:** non-cooperative equilibrium; random mutation module; performance discrimination module; multi-agent prediction and tracking



## 1. Introduction

In multi-agent intelligence, it would be common to encounter failed training, like the occurrence of non-convergence or low training speeds, if only individual agent learning methods were applied [1]. Multi-agent intelligence learning methods have been developed vigorously in recent years and amongst them the Multi-Agent Deep Deterministic Policy Gradient (MADDPG) method is widely studied as a basic algorithm since it shows excellent ability to export effective actions in multi-agent behaving problems [2]. For example, the MAGNet method [3] is one of the improved versions by perfecting the attention mechanism; the Decomposed Multi-Agent Deep Deterministic Policy Gradient (DE-MADDPG) method [4] is another improved version by coordinating local and global rewards to speed up convergence; the MiniMax Multi-agent Deep Deterministic Policy Gradient (M3DDPG) method [5], by promoting multi-agent adaptation to the environment; and the CAA-MADDPG [6], by increasing the attention mechanism. All these mentioned multi-agent intelligence learning methods inherit the cooperative learning strategies and show satisfying convergence but cannot achieve effective prediction and tracking when facing unfamiliar environments.

Some scholars attained upgraded predictive ability for their learning methods by introducing well-designed prediction framework. For example, Chien, JT and Hung, PY [7] combined prediction network and auxiliary replay memory in Deep Q Network (DQN) to estimate multiple states of the targets under different actions in order to support an advanced motion prediction. Zhu, PM and Dai, W [8] proposed a PER-MADDPG method

that uses information from multiple robots in the learning process to better predict the possible actions of the robot and thus to improve the learning efficiency. A deep neural network for evader pursuits by coordinating instant prediction and tracking was designed in [9]. Ure, NK and Yavas, MU [10] designed a novel learning framework for the prediction model used in Adaptive Cruise Control System, which enhances the situational awareness of the system by predicting the actions of surrounding drivers. In [11], a multi-agent observation and prediction system was established, which can realize the prediction and analysis of hurricane trajectory. Weng, XS [12] adopted the feature interaction technique from Graph Neural Networks (GNNs) to capture the way in which the agents interact with one another. The above studies show that for most multi-agent intelligence learning approaches based on cooperative learning strategy, it is feasible to reinforce predictive ability and to improve their learning efficiency by adding certain prediction network modules. However, the expected self-adaptation in unfamiliar environments had not been fulfilled yet.

Aiming for that, Wei, XL [13] improved the RDPG algorithm and achieved an increase in the action detection efficiency of the UAV together with radar stations under an unfamiliar battlefield environment by combining the MADDPG and RDPG methods. In [14], a new intrinsic motivation mechanism was introduced, Group Intrinsic Curiosity Module (GICM), into the MADDPG method. The GICM module encourages each agent to reach innovative and extraordinary states for a more comprehensive collaborative search of the current scenario in order to expand the adaptability of unfamiliar scenes. In [15], a parametric adaptive controller based on reinforcement learning was designed to improve the adaptability of the submarine in deep sea. Raziei, Z [16] developed and tested a Hyper-Actor Soft Actor-Critic (HASAC) deep reinforcement learning framework to enhance the adaptability of the agent when facing new tasks. Furuta, Y [17] designed an autonomous learning framework to ensure that the robot can adapt to different home environments. The above studies focus more on adaptability improvement and do not take promoting prediction and tracking ability to be the same priority. Therefore, the method's prediction and tracking ability cannot be reflected at the same time. The comparison of related research to this paper is shown in Table 1.

**Table 1.** Comparison of the work in this paper and related work.

| Method | Reference | Prediction | Tracking | Self-Adaption | Non-Cooperative |
| --- | --- | --- | --- | --- | --- |
| MADDPG | [3–6] | no | no | no | no |
| forecast | [7–12] | yes | no | no | no |
| adaptability | [13–17] | yes | no | yes | no |
| MPT-NCE | this paper | yes | yes | yes | yes |

In summary, most multi-agent learning approaches are based on cooperative learning strategies instead of non-cooperative equilibrium, and most of them lack the function of motion prediction and tracking. In order to achieve the prediction and tracking of the targets' action intention, as well as their self-adaptation to an unfamiliar environment simultaneously, a set of performance discrimination module and random mutation module is designed in this paper, and the non-cooperative strategy is applied, aiming to enhance the adaptability of the agents in unfamiliar environments.

This approach is named as the Multi-Agent Motion Prediction and Tracking method based on Non-Cooperative Equilibrium and is shortened as the MPT-NCE in the rest of this paper. By using the MPT-NCE, the agents can quickly identify unfamiliar environment information based on previous knowledge reserves, and are able to figure out the action intention of their counterparts and then track them in the game antagonism environment. Compared with the MADDPG learning method, the proposed MPT-NCE has a faster convergence speed and better performance. The remaining sections are arranged as follows: Section 2 is about the related work, in which the MADDPG multi-agent intelligence learning method is introduced briefly. Section 3 presents the details of the MPT-NCE, in which the

random mutation module and performance discrimination module are designed to realize the motion prediction and tracking of the agents. Section 4 provides the performance of our method in two different experiments to evaluate its effectiveness. Discussion and summary are presented at last in Section 5.

## 2. Related Work

This section introduces the MADDPG multi-agent intelligence learning method in basic terms.

### 2.1. Multi-Agent Deep Deterministic Policy Gradient Learning Network

The MADDPG is a deep learning method based on AC (Actor-Critic) architecture. For any intelligent individual or group, there is a policy network and an evaluation network corresponding to its features [18]. At moment $t$, the policy network generates a continuous action $A_t$ based on the state $S_t$ in the environment. After that the environment interacts with $A_t$ to generate the state $S_{t+1}$ at the next moment. The critic network will get the evaluation value $Q$ of the decision process of the policy network at moment $t$ based on $S_t$ and $A_t$. The structure of the critic network of MADDPG is similar to that of the policy network, where the input is the state $S$ and the action $A$ generated by the policy network, and the output is the determined $Q$ value [19].

### 2.2. Multi-Agent Deep Deterministic Policy Gradient Learning Method

The strategy parameters for each agent in the MADDPG method are $\theta = (\theta_1, \theta_2, \ldots, \theta_N)$ and the joint strategy is $H : \pi = (\pi(\theta_1), \pi(\theta_2), \ldots, \pi(\theta_N))$.

The multi-agent interactive learning experience replay set [20] is

$$S_t, a_1^t, a_2^t, \ldots, a_N^t, r_1^t, r_2^t, \ldots, r_N^t, S_{t+1} \tag{1}$$

where $S_t = (s_1^t, s_2^t, \ldots, s_N^t)$ is the set of states of an agent at moment $t$, $A_t = (a_1^t, a_2^t, \ldots, a_N^t)$ is the set of actions of an agent at moment $t$, $R_t = (r_1^t, r_2^t, \ldots, r_N^t)$ is the set of rewards obtained after the action interacts with the environment at moment $t$, and $S_{t+1} = (s_1^{t+1}, s_2^{t+1}, \ldots, s_N^{t+1})$ is the set of states of the agent at moment $t + 1$ [21].

The work of the critic network of MADDPG requires the participation of the states and actions of all agents, and the evaluation is updated by observing the states and actions of other agents to achieve interactive learning. The loss function of the critic network [22] is

$$L = \frac{1}{K} \sum_{t=1}^{K} (y_t - Q(S_t, A_t, \theta^Q))^2 \tag{2}$$

where, $K$ is the total number of agents in the environment, $\theta^Q$ is the critic network parameter, and $y_t$ is the target $Q$ value of the target critic network output, whose formula is shown in (3) [23]. $Q(S_t, A_t, \theta^Q)$ is the $Q$ value of critic network output, which represents the optimal value of performing action $A_t$ in the current state $S_t$. The target $Q$ value and the $Q$ value of critic network output together update the critic network parameters in advance for the next moment.

$$y_t = R_t + \gamma Q'(S_{t+1}, A_{t+1}, \theta^{Q'}) \tag{3}$$

where $A_t = \pi_{\theta^u}(S_t)$, $\pi_{\theta^u}$ is the strategy in policy network which can output actions $A_t$ according to status $S_t$, and $\gamma$ is the discount factor.

The learning objective of the critic network is to minimize $L(\theta^Q)$, and the parameters $\theta^Q$ of the critic network are updated by the gradient descent method.

The gradient calculation formula of policy network [24] is

$$\nabla_{\theta^u} J = \frac{1}{K} \sum_{t=1}^{K} \nabla_{\theta^u} \pi(S_t, \theta^u) \nabla Q(S_t, A_t, \theta^Q) \tag{4}$$

where $\theta^u$ is the policy network parameter. In Equation (4) above, $A_t$ is the output of the policy network, and other parameters are obtained by experience replay set. The policy network aims to maximize the output of the critic network, so the parameters of the policy network can be updated by the gradient ascent method $\theta^u$ [18].

Since the MADDPG learning method has no prediction framework, it is difficult for the MADDPG learning method to achieve prediction learning and intelligent tracking when targeting complex and unfamiliar environments. In this paper, the shortcomings of MADDPG method in unfamiliar environments has been solved, a performance discrimination module and a random mutation module have been designed to realize each other's prediction and tracking in unfamiliar environments.

## 3. System Design

This section is divided into two main parts. Firstly, it provides an introduction to the composition of the system, and then it explains the random mutation module and performance discrimination module included in the system.

### 3.1. System Composition

Figure 1 shows the composition of the MPT-NCE system, which overcomes the shortcomings of the MADDPG method that does not adapt to unfamiliar environments. A random mutation module and a performance discrimination module have been designed for the agents to achieve the prediction and tracking of opposite agents' motion in unfamiliar environments.

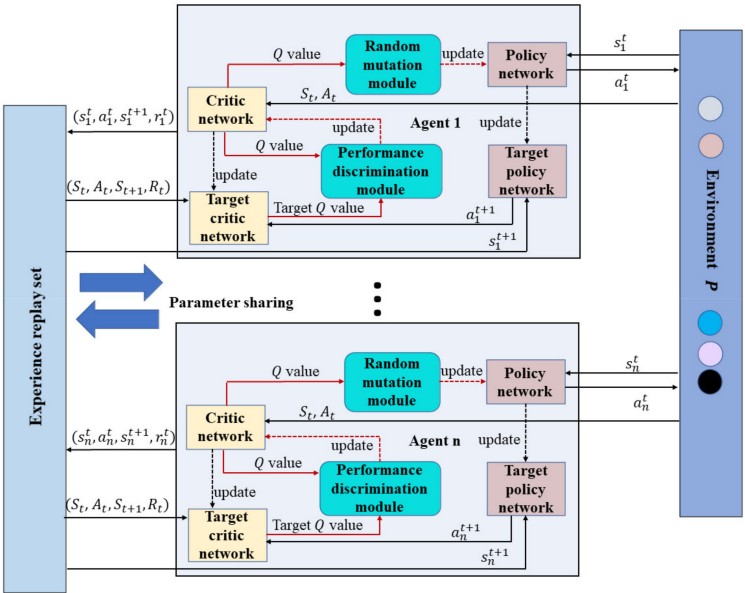

**Figure 1.** MPT-NCE system.

The MPT-NCE system shown in Figure 1 involves all the agents at time $t$, the experience replay set, and the environment $P$. Among them, the experience replay set stores the interaction information of each agent with the environment, which is shared with all the agents' parameters. The environment $P$ contains hidden areas, obstacle areas, and different classes of agents. A performance discrimination module and a random mutation module are designed for each agent. The policy network of each agent gets the action based on state $s_i^t$ as $a_i^t = \pi_\theta(s_i^t)$, and its action $a_i^t$ interacts with environment $P$ to get a new state $s_i^{t+1}$ with reward $r_i^t$. What follows is that the set of state actions and rewards of all agents $\{S_t, a_1^t, a_2^t, \ldots, a_N^t, r_1^t, r_2^t, \ldots, r_N^t, S_{t+1}\}$ is deposited into the experience replay set $D$. The agent policy network recommends an action using a non-cooperative policy and considers only its own state. The target policy network of the agent is responsible for predicting the

action $a_i^{t+1}$ at time $t + 1$, and the target critic network uses this action to predict the target $Q$ value at $t + 1$ to participate in the performance discrimination module.

The performance discrimination module introduces a time difference concept into the loss function to guide the update direction of the critic network. The output $Q$ value of the critic network and the target $Q$ value of the target critic network are the inputs of the performance discrimination module. Since the target $Q$ value contains the prediction information at time $t + 1$, the performance discrimination module differentiates it from the output $Q$ value at time $t$ to achieve the prediction of the behavioral intention of the opponent in an unfamiliar environment.

The random mutation module adds the information of other agents to guide the policy network to give an output on the basis of prediction. The output $Q$ value of the critic network updates the policy network after the random mutation module to realize the learning tracking of the other agents' behavior in unfamiliar environments. Compared with the MADDPG method, the non-cooperative equilibrium multi-agent prediction and tracking method has better environmental self-adaptation and can realize the prediction and tracking of each other's action intention in an unfamiliar environment.

*3.2. Module Design*

3.2.1. Performance Discrimination Module

In order to achieve the effect of model prediction, differential reinforcement learning method is introduced in the performance discrimination module. The state function can be expressed as

$$V^\pi(s) = E[G_t | S_t = s] \tag{5}$$

$G_t$ in Equation (5) is the state reward, which can be approximated by $R_{t+1} + \lambda V^\pi(S_{t+1})$. Combining with Equation (5), $R_{t+1} + \lambda V^\pi(S_{t+1}) - V^\pi(S_t)$ is referred to as the time difference error value.

The process of using this approximation method to replace the sum of rewards and punishments of cumulative discounts is a guiding process. Therefore, the predictive learning problem can be solved only according to two consecutive states and corresponding reward and punishment, updating the state function based on the time difference error values.

$$V^\pi(S_{t+1}) = V^\pi(S_t) + \alpha[R_{t+1} + \lambda V^\pi(S_{t+1}) - V^\pi(S_t)] \tag{6}$$

Equation (6) forms a state function immediately at moment $t + 1$ and is updated using the instantaneous reward and punishment values observed from the environment and the estimated state value function $V^\pi(S_{t+1})$ for $t + 1$. The state value function can be approximated by the $Q$ value.

The critic network is able to observe the states and actions of all the agents, so the performance discrimination module guides the update direction of the critic network. The performance discrimination module incorporates the idea of time difference error, where the $Q'$ value of the next moment estimated by the target critic network is differenced from the $Q$ value observed by the critic network from the environment to achieve state prediction. The output $Q$ value and the target $Q$ value are substituted into the time difference error value in Equation (6) to form the loss function.

$$F = \frac{1}{K} \sum_{t=1}^{K} (R_t + \alpha Q'(S_{t+1}, A_{t+1}, \theta^{Q'}) + \delta Q'(s_{t+1}, a_{t+1}, \theta^{Q'}) - Q(S_t, A_t, \theta^Q))^2 \tag{7}$$

where $\alpha$ is the discriminant parameter and $\delta$ is the evaluation parameter. $Q'(S_{t+1}, A_{t+1}, \theta^{Q'})$ is the evaluation $Q'$ value of the target critic network based on the predicted action and state at moment $t + 1$. $Q'(S_{t+1}, A_{t+1}, \theta^{Q'})$ is the value output based on the states and actions of all agents. $Q'(s_{t+1}, a_{t+1}, \theta^{Q'})$ is the value output based on the state and action of agent of itself. $Q(s_t, a_t, \theta^Q)$ is the corresponding evaluation $Q$ value of the agent's own state and action at moment $t$.

Achieving the prediction of the opponent's behavioral intention requires not only predicting the information of other agents at moment $t + 1$, but also ensuring the importance of its own moment $t + 1$ to make the agent choose the correct strategy. Therefore, the $Q'$ value of the agent's own state at moment $t + 1$ is introduced in Equation (7) to highlight the importance of its own action. The weights of $Q'(S_{t+1}, A_{t+1}, \theta^{Q'})$ and $Q'(s_{t+1}, a_{t+1}, \theta^{Q'})$ are controlled by $\alpha$ and $\delta$ to ensure the parameter balance of the difference formula. The predicted $Q'$ value at moment $t + 1$ and the $Q$ value at moment $t$ do the difference to guide the update direction of the critic network at the next moment to achieve the effect of model prediction. The target critic network outputs overestimated $Q'$ values that can produce local optima, and the discriminant parameter $\alpha$ and the evaluation parameter $\delta$ can also intervene to attenuate the $Q'$ values to avoid this phenomenon. To ensure the convergence of the critic network, a gradient descent update method is used.

$$\theta_{t+1}^Q = \theta_t^Q - \eta \frac{\partial}{\partial \theta^Q} F(\theta_t^Q) \tag{8}$$

The performance discrimination module ensures the predictive ability of MPT-NCE methods for opposing agents in unfamiliar environments by making a time difference between the $Q'$ values and the $Q$ values of the agents.

The working flow chart of the performance discrimination module is shown in Figure 2. Each interaction between an agent and the environment is recorded as $\{S_t, a_1^t, a_2^t, \ldots, a_N^t, r_1^t, r_2^t, \ldots, r_N^t, S_{t+1}\}$ which then is deposited into the experience replay set. The target critic network collects the information from the experience replay set to output the target $Q$ value, which is then given to the performance discrimination module together with the $Q$ value that is given by the critic network. After that, the performance discrimination module calculates the loss and updates the critic network instantly using the gradient descent method until the maximum training number is reached. The performance discrimination module takes the concept of time difference to ensure the predictive ability of the method in unfamiliar environments.

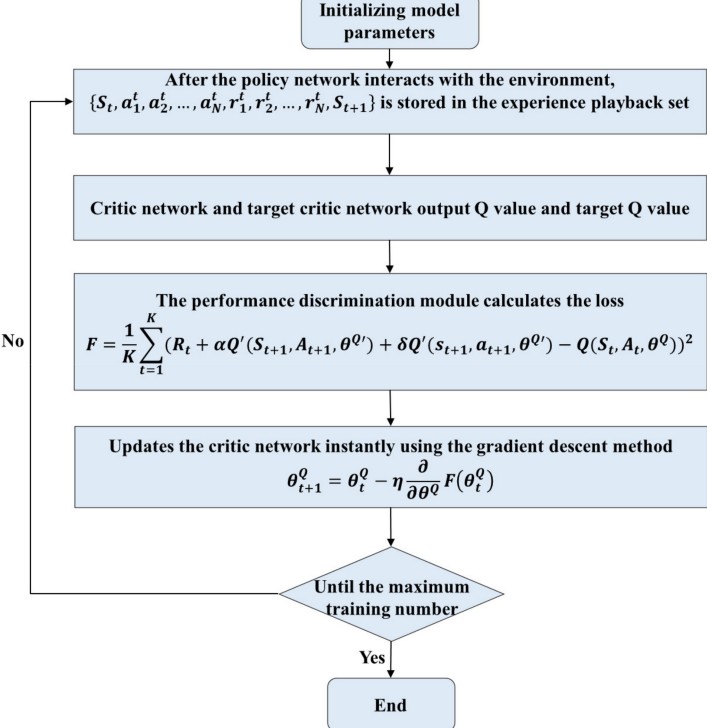

**Figure 2.** Working flow chart of the performance discrimination module.

### 3.2.2. Random Mutation Module

The output of the model is produced by the policy network, and a random mutation module is introduced for the update of the policy network in order to achieve the effect of tracking and learning from unfamiliar variant sources. The critic network combines the state and action information of other agents to output the evaluation value, and then to guide the update of the policy network. Based on that, the value $Q'$ is introduced to represent the predicted state and action of other agents in the next moment, so that to ensure the updating direction of the policy network and to achieve the effect of tracking and learning for unfamiliar agents.

$$\nabla_{\theta^u} T = \frac{1}{K} \sum_{t=1}^{K} \nabla_{\theta^u} \pi(S_t, \theta^u) \nabla(\beta Q(S_t, A_t, \theta^Q) + \varphi Q'(S_{t+1}, A_{t+1}, \theta^{Q'})) \tag{9}$$

where $K$ is the total number of agents, $\beta$ is the mutation coefficient, and $\varphi$ is the target mutation coefficient.

The random mutation module should introduce the prediction information of other agents at moment $t + 1$, which will ensure that agent can realize the tracking of the opponents' behavior based on prediction learning. Therefore, the $Q'$ value is introduced in Equation (9) to include the prediction of the next moment of the other agent's behavioral action, which ensures the adaptability of the method to the unfamiliar environment and assists the policy network as a guidance to accomplish updating. $\beta$ and $\varphi$ control the weights of the $Q$ value and $Q'$ value to ensure the balance of the output of the random mutation module. The mutation coefficient $\beta$ retains the main guiding role of $Q$ values for policy network updates and also serves as an attenuation for over-estimated $Q$ values. The target mutation coefficient $\varphi$ intervenes to attenuate the introduction of overestimated $Q'$ values while introducing $Q'$ values at moment $t + 1$ to avoid reaching a local optimum during training. Equation (10) is the gradient formula of the random mutation module for the update of the policy network.

$$\theta_{t+1}^u = \theta_t^u + \eta \frac{\partial}{\partial \theta^u} T(\theta_t^u) \tag{10}$$

The working flow chart of the random mutation module is shown in Figure 3. As long as the interaction between the agent and the environment happens, the critic network would output $Q$ values based on the current interaction information and give it to the random mutation module. After that, the random mutation module will guide the update direction of the policy network and update the policy network using the gradient ascending method until the maximum training number is reached. The random mutation module adds the predicted states and actions of other agents in the next moment to ensure the update direction of the policy network and realize the tracking learning of each other in the unfamiliar environment.

The performance discrimination module possesses the prediction learning ability for other agents, and the random mutation module guides the updated output of the policy network, so that the MPT-NCE method achieves the tracking of each other's behavioral intention while ensuring the prediction learning. The pseudo-code of the MPT-NCE method is shown in Table 2.

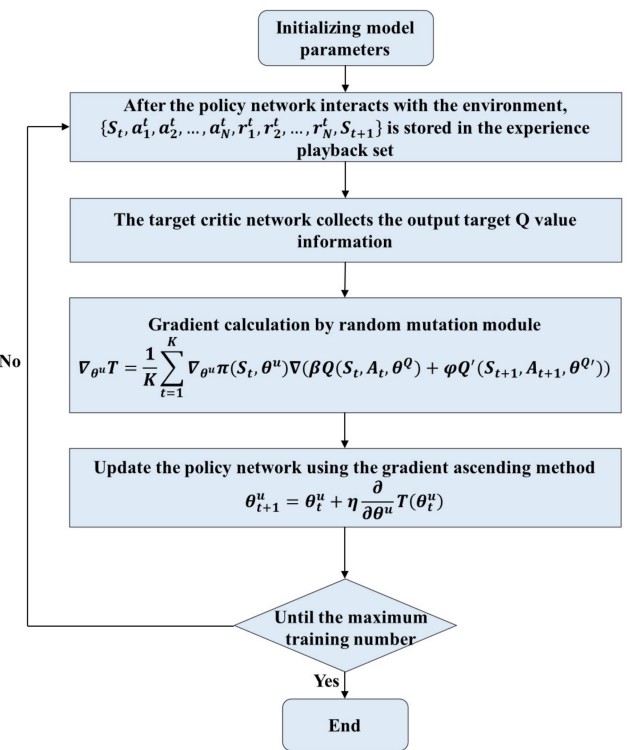

**Figure 3.** Working flow chart of random mutation module.

**Table 2.** The pseudo-code of the MPT-NCE method.

| Pseudo-Code |
| --- |

1. Randomly initialize the network parameters $\theta^u$, $\theta^Q$, $\theta^{u\prime} = \theta^u$, $\theta^{Q\prime} = \theta^Q$, clear the experience replay set
2. For I from 1 to N, iterating

    (a) Initialize state $S$.

    (b) Each agent, gets $A_t = \pi_\theta(S_t)$ in the policy network based on state $S_t$ at moment $t$.

    (c) New state $S_{t+1}$ and reward $R_t$ are obtained by action $A_t$, and store $\left\{s_t, a_1^t, a_2^t, \ldots, a_N^t, r_1^t, r_2^t, \ldots, r_N^t, s_{t+1}\right\}$ into experience replay set $D$.

    (d) The experience replay set outputs $K$ samples for each agent to updating the network.

    (1) The target policy network outputs the optimal action $A_{t+1} = \pi_\theta(S_{t+1})$ at the next moment.

    (2) The $Q$ values are computed by the target critic network with state and action as inputs and $y_t = R_t + \gamma Q'(S_{t+1}, A_{t+1}, \theta^{Q\prime})$ as outputs.

    (3) The performance discrimination module is based on $F = \frac{1}{K}\sum_{t=1}^{K}(R_t + \alpha Q'(S_{t+1}, A_{t+1}, \theta^{Q\prime}) + \delta Q'(s_{t+1}, a_{t+1}, \theta^{Q\prime}) - Q(S_t, A_t, \theta^Q))^2$ as the loss function to update the critic network.

    (4) The random mutation module updates the policy network by $\nabla_{\theta^u} T = \frac{1}{K}\sum_{t=1}^{K} \nabla_{\theta^u} \pi(s_t, \theta^u)\nabla(\beta Q(s_t, a_1, a_2, \ldots, a_N, \theta^Q) + \varphi Q'(S_{t+1}, A_{t+1}, \theta^{Q\prime}))$

    (5) The target critic network and target policy network parameters are updated at a certain frequency.

$$\theta^{Q\prime} = \tau\theta^Q + (1-\tau)\theta^{Q\prime}$$

$$\theta^{u\prime} = \tau\theta^u + (1-\tau)\theta^{u\prime} \qquad \tau \text{ is the update coefficient}$$

End for

## 4. Experiments and Results

### 4.1. Experimental Environment

In order to verify the effectiveness of the MPT-NCE, two experiments were designed and conducted in this section. The experimental conditions are different in the number of agents and the complexity of the environments. For illustration, blue spheres are used to represent the detection agents, red spheres represent the interference agents, and purple spheres represent the mutable interference agents. The mutated agent is labeled by purple sphere with expanded size.

$$reward_d = -0.1 \times \sum_1^{n_i} \sqrt{(x_d - x_{i_{n_i}})^2 + (y_d - y_{i_{n_i}})^2} \tag{11}$$

$$reward_i = 0.9 \times \sum_1^{n_d} \sqrt{(x_{d_{n_d}} - x_i)^2 + (y_{d_{n_d}} - y_i)^2} \tag{12}$$

Equation (11) is the reward formula for detection agents, where $n_i$ is the number of interference agents in the experiment, $i_{n_i}$ is all interfering agents, and $x$ and $y$ are the horizontal and vertical coordinates where the agent locates. Equation (12) is the reward formula for the interference agents, where $n_d$ is the number of detection agents in the experiment, and $d_{n_d}$ is all detection agents. The reward of the detection agents increases when the distance between the detection agents and the interference agents becomes smaller. Similarly, the reward of the detection agents decreases when the distance between the detection agents and the interference agents becomes larger. The detection agents collide with the interference agents, indicating that the detections are effective. The interference agents maintain the distance with the detection agents can get reward, and if the purple interference agents are successfully detected, it will produce random mutation, the purple balls will become larger schematically. Stability and convergence are the Experimental Evaluation Index. This article sets the steady state of the curve as the curve's final reward fluctuations to not exceed 10% of the overall value. Additionally, the episode to achieve stable reward can reflect the convergence, and the convergence is better if the episode required is smaller.

The performance discrimination module guides the critic network update through the gradient descent method. The usage of the $Q'$ value of the agent itself in the loss function can enhance the prediction ability of the MPT-NCE method, so that the reward curve has better convergence. The random mutation module guides the policy network update through the gradient ascent method with the aim of obtaining a larger $Q$ value with $Q'$ value, where the $Q$ value is an assessment value according to the agent's state and action, and is proportionate to the increased current rewards of the agent.

In summary, the fast convergence speed and high reward values can reflect the effectiveness of the MPT-NCE method. In the prediction experiments, the convergence speed of the reward curves can reflect the prediction ability of the MPT-NCE method, and the curve reaches stability indicating that the prediction of the opposing agent's behavior is achieved. In the tracking experiments, the reward values of the reward curves reflect the tracking ability of the MPT-NCE method, and the higher reward values indicate the closer the distance between the blue detection agents and the interference source.

The experimental system uses Ubuntu 18.04 and the processor is Intel(R) i7-10875H. In addition, the experimental code is based on the parl architecture. The other used parameters are set as in Table 3.

**Table 3.** Experimental parameter.

| Parameter Type | Setting Value | Parameter Type | Setting Value |
|:---:|:---:|:---:|:---:|
| $\theta^u$ | 0.5 | $\theta^Q$ | 0.5 |
| $\alpha$ | 0.8 | $\beta$ | 0.8 |
| $\delta$ | 0.2 | $\varphi$ | 0.2 |
| $\gamma$ | 3-$e$ | $\tau$ | 0.01 |
| training duration | 5 | Maximum training times | 25,000 |
| Acceleration | 3.5–4.5 | Scene size | $1 \times 1$ |

*4.2. Prediction Experiment*

The prediction experiment contains 4 blue detection agents, 4 red interference agents, 2 gray hidden areas, and 2 black obstacle areas. All agents are entities, and agents with entity properties will bounce off after collision. After the agents enter the hidden area, the agents outside the hidden area cannot obtain its position. The black obstacle areas are impassable areas, in which the passage of any agent is prevented.

The process of prediction experiment is shown in Figure 4. To test the prediction ability of the blue detection agents, the active area of the red interference agents is limited to the red dashed box. As shown in Figure 4a, the red interference agents are targets for the blue detection agents whose task is to detect the red interference agents. At the early stage of training, the blue detection agents have no predictive capability and all appear outside the dotted box. As shown in Figure 4b, as the training times increases, some of the blue detection agents soon move into the red dotted box. This phenomenon indicates that the behavior of the blue detection agents is not a unified decision, which reflects the non-cooperative decision of the MPT-NCE method. As shown in Figure 4c, after training, the experimental scenario was re-demonstrated, and all the blue detection agents had moved into the red dotted box, indicating that the blue detection agents all have the ability to predict the action pattern of the red interference agents. The results of the prediction experiment show that the trained blue detection agents can predict the behavior pattern of the red interference agents. The time difference function of the performance discrimination module guides the update of the critic network so that the blue detection agents have the prediction function. In order to reflect the effectiveness of the prediction results, this paper adopts a binomial distribution test for the prediction rate, randomly selecting 100 scenes for observation, using 1 to indicate the success of the detection monomer prediction and 0 to indicate the failure of the prediction. The result shows that the number of 1 is 92, which proves that the prediction rate of the MPT-NCE method reaches 90%.

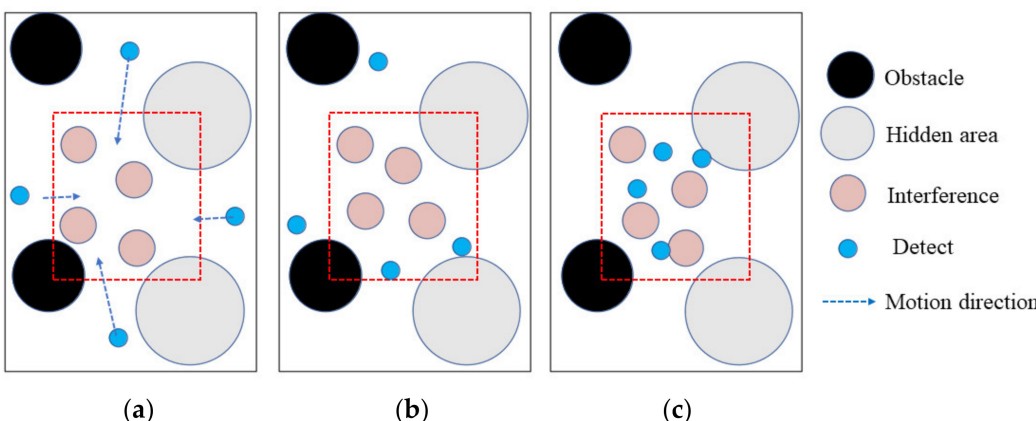

**Figure 4.** Prediction experiment: (**a**) shows the movement trend in the pre-training period of detection agents, (**b**) shows the effect in the middle of the training period, and (**c**) shows the action results after training.

Figure 5 shows the rewards in the prediction experiment. In Figure 5a, the multi-agent rewards during training increases significantly, indicating that the random mutation module guides the $Q$ value to increase, which proves the effectiveness of the MPT-NCE method.

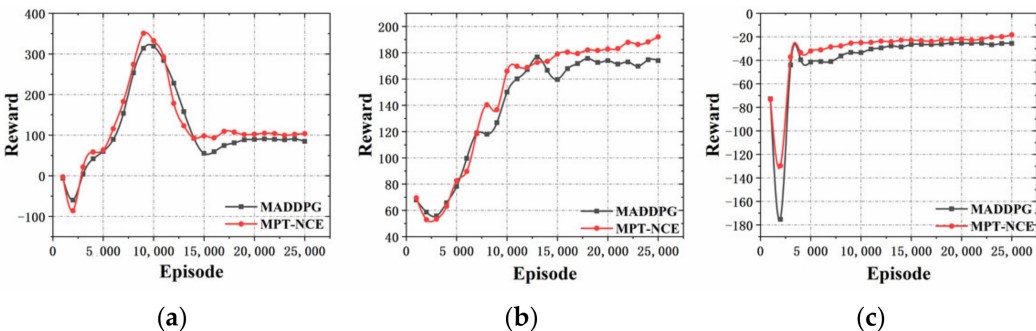

(a)       (b)       (c)

**Figure 5.** Reward curves in prediction experiment: (**a**) shows the comparison chart of the reward curve between the MPT-NCE method and the MADDPG method. (**b**) shows the blue detection agents' reward curves of MPT-NCE method and MADDPG method, respectively. (**c**) shows the red interference agents' reward curves of MPT-NCE method and MADDPG method, respectively.

The reward curve becomes steady latterly, which indicates that the used policy network works as expected, the prediction of the counterparts' behavior in unfamiliar environment is realized, and the non-cooperative equilibrium has been reached. The MPT-NCE method has faster convergence speed and higher value of the reward function compared with the MADDPG method, which indicates that MPT-NCE method has much better prediction ability and effect. In Figure 5b,c, the agents evolve independently while the non-cooperative distribution strategy is carried out during the training of the blue and red agents. The reward function curves of the both sides finally reach the equilibrium steady state, indicating that the performance discrimination module improves the convergence speed of the method and proves the effectiveness of the MPT-NCE method.

In order to further verify the prediction effect of the proposed method, the training times and the value of the rewards are evaluated and compared with that of the the MADDPG method. Figure 6 illustrates the comparason in detail, all the presented results in Figure 6 are obtained when the accelerations of the blue detection agents and the red interference agents are both set to be 4.

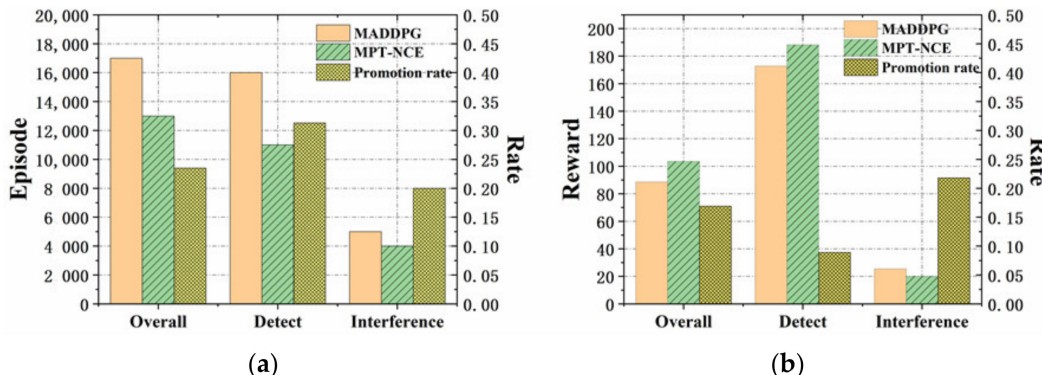

(a)       (b)

**Figure 6.** Performance comparison chart of prediction experiment: (**a**) shows a comparison chart of training times and (**b**) shows a comparison chart of the reward function values.

As shown in Figure 6a, the training times of MPT-NCE method are compared with that of MADDPG method. It is obvious that the MPT-NCE method reaches stability in a shorter time, and the convergence rate is increased by 23.52% compared with MADDPG method. As shown in Figure 6b, the comparison of the reward function values of MPT-NCE

method and MADDPG method shows that the reward of the MPT-NCE method is larger, which is improved by 16.89% compared with the MADDPG method.

### 4.3. Tracking Experiment

The tracking experiment contains 4 blue detection agents, 1 red interference agent, 2 purple mutable interference agents, 2 gray hidden areas, and 2 black obstacle areas. The purple mutable interference agents will mutate with the detection of the detection agents.

Tracking experiment 1 is illustrated by Figure 7. After training, all agents can clear the target and complete the task. In Figure 7a, the targets of blue detection agents are the red ones and the purple interference agents, and their task is to detect the targets. The targets of red and purple interference agents are the blue detection agents, and their task is to increase the distance to the blue detection agents. As shown in Figure 7b, each agent acts according to the task target. As shown in Figure 7c, the blue detection agents are able to track the red and purple interference agents, while the red and purple interference agents try to increase the distance from the blue detection agents.

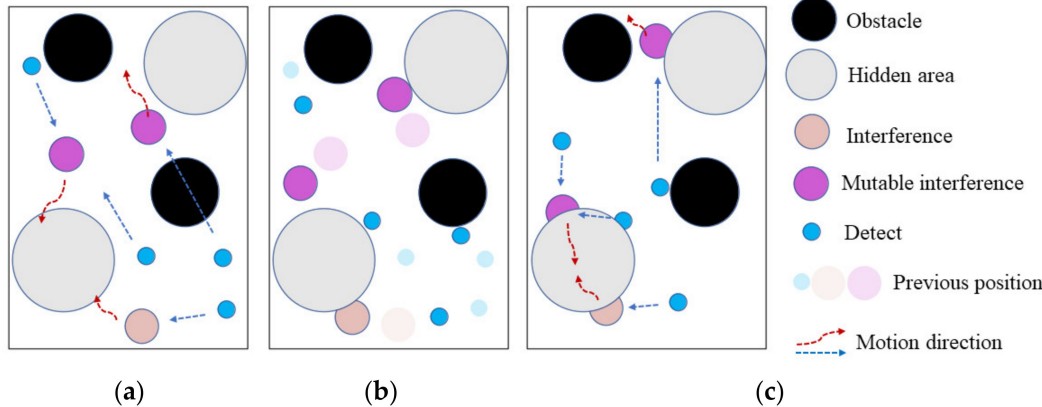

**Figure 7.** Tracking experiment before purple interference agents' mutation (tracking experiment 1): (**a**) shows the motion trend of each agent, (**b**) shows the action of each agent, and (**c**) shows the results after each agent action.

Tracking experiment 2 is illustrated by Figure 8, when the purple interference agent is successfully detected, the volume variation increases significantly. In Figure 8a, it can be seen that the blue detection agent targets are red and purple mutable interference agents. As shown in Figure 8b, each agent acts according to the task target. As shown in Figure 8c, it can be seen that the blue detection agents can track the mutated purple interference agents. The tracking experiment results show that the trained blue detection agents can effectively track the mutated purple interference agents. The random mutation module can influence the output of the policy network on the basis of predictive learning, and realize the behavior tracking of the other party in unfamiliar environment. It reflects the self-adaptation of the MPT-NCE method in unfamiliar environment.

In the tracking experiments, the reward value reflects the tracking ability of the MPT-NCE method. The closer the distance between the blue detection agent and the interference source, the higher the reward value.

Figure 9 shows the curve of the reward in the tracking experiment. In Figure 9a, the multi-agent reward of the MPT-NCE method rises significantly after training, and has a faster convergence rate with higher reward function compared with the MADDPG method, which proves the effectiveness of the MPT-NCE method. In Figure 9b,c, when blue detection agents and red interference agents are trained, the evolution process of each agent is independent of each other, and the non-cooperative distribution strategy is implemented. The reward function curves both reach the equilibrium steady state, indicating that the experimental results reach the non-cooperative equilibrium.

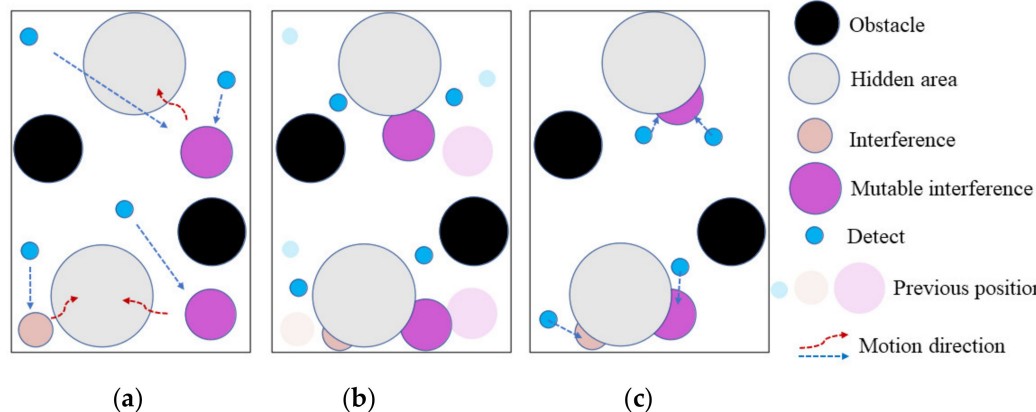

**Figure 8.** Tracking experiment after mutation of purple interference agents (tracking experiment 2): (**a**) shows the motion trend of each agent, (**b**) shows the action of each agent, and (**c**) shows the results after each agent action.

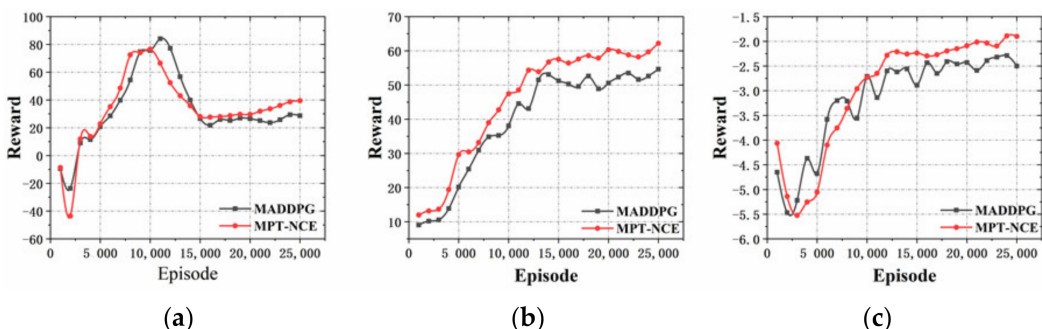

**Figure 9.** Reward function curve of tracking experiment: (**a**) shows the comparison chart of reward curve between the MPT-NCE method and MADDPG method, (**b**) shows the MPT-NCE method and MADDPG method of blue detection agents reward function curve comparison diagram, and (**c**) shows the MPT-NCE method and MADDPG method of red interference agents reward function curve comparison diagram.

To further verify the tracking effect of the MPT-NCE method compared with the MAD-DPG method, the training times and rewards are checked, using the approach introduced previously in Section 4.2. The acceleration of all the agents including the blue detection ones, the red, or purple interference ones were set to be the same.

As shown in Figure 10a, the training times of MPT-NCE method and MADDPG method are compared. It is obvious that MPT-NCE method is stable in a shorter time and the convergence rate is increased by 11.76% compared to MADDPG method. Figure 10b shows the comparison of the reward function values of MPT-NCE method and MADDPG method. The reward function value of the MPT-NCE method is larger and the tracking performance is improved by 25.85% compared to the MADDPG method.

### 4.4. Discussion of Experimental Results

The results of the prediction and tracking experiments are summarized in Tables 4 and 5.

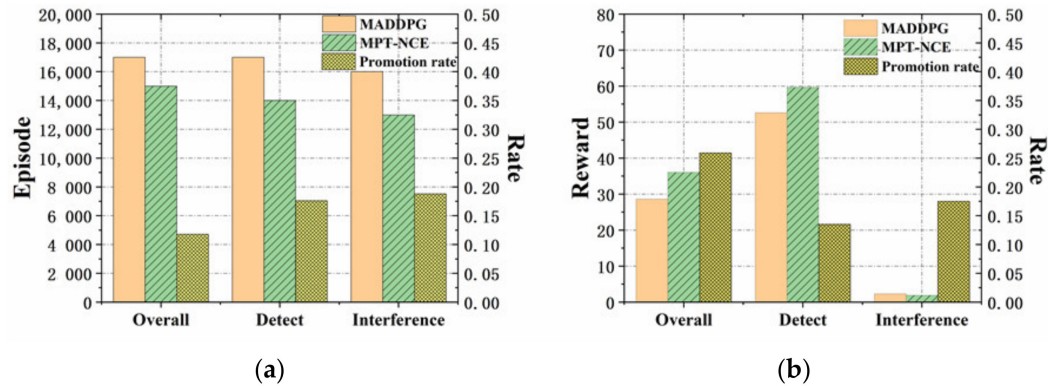

**Figure 10.** Tracking experimental performance comparison chart: (**a**) shows the comparison chart of training times and (**b**) shows the comparison chart of reward function values.

**Table 4.** Comparison of stable reward values under different experimental environments.

| Experimental Environment | Method | Curves | Stable Reward Value | Percentage Increase |
|---|---|---|---|---|
| prediction | MADDPG | Overall | 88.6 | 16.89% |
| prediction | MPT-NCE | Overall | 103.56 | |
| prediction | MADDPG | Detection | 172.97 | 8.85% |
| prediction | MPT-NCE | Detection | 188.27 | |
| prediction | MADDPG | Interference | −25.45 | 21.77% |
| prediction | MPT-NCE | Interference | −19.91 | |
| tracking | MADDPG | Overall | 28.63 | 25.85% |
| tracking | MPT-NCE | Overall | 36.03 | |
| tracking | MADDPG | Detection | 52.6 | 13.46% |
| tracking | MPT-NCE | Detection | 59.68 | |
| tracking | MADDPG | Interference | −2.29 | 17.47% |
| tracking | MPT-NCE | Interference | −1.89 | |

**Table 5.** Comparison of stable training times under different experimental environments.

| Experimental Environment | Method | Curves | Stable Training Times | Percentage Increase |
|---|---|---|---|---|
| prediction | MADDPG | Overall | 17,000 | 23.52% |
| prediction | MPT-NCE | Overall | 13,000 | |
| prediction | MADDPG | Detection | 16,000 | 31.25% |
| prediction | MPT-NCE | Detection | 11,000 | |
| prediction | MADDPG | Interference | 5000 | 20% |
| prediction | MPT-NCE | Interference | 4000 | |
| tracking | MADDPG | Overall | 17,000 | 11.76% |
| tracking | MPT-NCE | Overall | 15,000 | |
| tracking | MADDPG | Detection | 17,000 | 17.64% |
| tracking | MPT-NCE | Detection | 14,000 | |
| tracking | MADDPG | Interference | 16,000 | 18.75% |
| tracking | MPT-NCE | Interference | 13,000 | |

Table 4 shows the comparison of the stable reward values under different experimental environments. The rewards after training by the MPT-NCE method are all significantly improved compared to the MADDPG method. In the prediction experiments, the overall, detection and interference reward curves after training with the MPT-NCE method are improved by 16.89%, 8.85%, and 21.77%, respectively, compared with the MADDPG method. In addition, in the tracking experiment, the overall, detection and interference reward curves of the MPT-NCE method after training are improved by 25.86%, 13.46%, and 17.47%, respectively, compared with those of the MADDPG method.

Table 5 shows the comparison of the stable training times under different experimental environments. The training speed of the MPT-NCE method is significantly improved compared to the MADDPG method. In the prediction experiments, the convergence speed of the overall, detection, and interference curves of the MPT-NCE method after training are improved by 23.52%, 31.25%, and 20%, respectively, compared with the MADDPG method. In addition, in the tracking experiment, the convergence speed of the overall, detection, and interference curves of the MPT-NCE method after training are improved by 11.76%, 17.64%, and 18.75%, respectively, compared with those of the MADDPG method.

In summary, in the prediction experiment, the convergence speed of the curve can reflect the prediction ability of the method, and the prediction ability of MPT-NCE method is 23.52% higher than that of MADDPG method. In the tracking experiment, the reward after curve stability can reflect the tracking performance of the method, and the tracking ability of MPT-NCE method is 25.85% higher than that of MADDPG method. Compared with the MADDPG method, the MPT-NCE method proposed in this paper has better prediction and tracking performance, so it can be applied to the training scenarios of multi-agent prediction and tracking.

## 5. Conclusions

In this paper, a Multi-Agent Motion Prediction and Tracking method based on non-cooperative equilibrium (MPT-NCE) is proposed. Taking the MADDPG method as foundation, a set of novel performance discrimination module and random mutation module is designed, aiming for quick identification and great adaptability to strange environments, and also leading to the realization of motion prediction and tracking of the counterparts in the game confrontation environment. The performance discrimination module using the time difference function guides the critic network to update and improve the prediction ability of the method. The random mutation module guides the update of policy network on the basis of predictive learning, and realizes the tracking of opposite agents' behavior intention. The proposed MPT-NCE method can be applied to various multi-target pursuit scenarios, and of high potential to solve many problems such as target prediction and task pursuit in other systems. In addition, future research will focus on solving prediction tracking problems in dynamic environments and integrating MPT-NCE with other systems.

In order to verify the multi-agent prediction and tracking ability of the proposed method, two groups of multi-agent experiments were conducted. The results of the prediction experiment show that compared with the MADDPG method, the prediction ability of the MPT-NCE method is improved by 23.52%, the performance is improved by 16.89%, and the prediction rate is more than 90%. The results of tracking experiment show that the MPT-NCE method improves the convergence rate by 11.76% and the tracking ability by 25.85% compared with the MADDPG method. The experimental results effectively prove that the MPT-NCE method has superior environmental adaptability and prediction tracking ability.

**Author Contributions:** Conceptualization, Y.L. and M.Z.; methodology, Y.L., M.Z. and S.W.; software, M.Z.; validation, Y.L., M.Z. and H.Z.; formal analysis, Y.L. and S.W.; investigation, M.Z. and H.Z.; resources, Y.L. and S.W.; data curation, M.Z.; writing—original draft preparation, M.Z. and S.W.; writing—review and editing, M.Z., Y.L. and S.W.; visualization, H.Z. and Y.Q.; supervision, H.Z. and Y.Q.; project administration, S.W. and Y.Q.; funding acquisition, Y.L. All authors have read and agreed to the published version of the manuscript.

**Funding:** This research was supported by the National Natural Science Foundation of China with grant Nos. 61973308, 62003350, and 62175258, also the Fundamental Research Funds for the Central Universities of China.

**Institutional Review Board Statement:** Not applicable.

**Informed Consent Statement:** Not applicable.

**Data Availability Statement:** The data presented in this study are available upon reasonable request from the corresponding author.

**Conflicts of Interest:** The authors declare no conflict of interest.

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
