# Peer review of "A Multi-Agent Motion Prediction and Tracking Method Based on Non-Cooperative Equilibrium"

_mathematics, doi:10.3390/math10010164_

Round 1

Reviewer 1 Report

This manuscript presents the “A Multi-agent Motion Prediction and Tracking method based on non-cooperative equilibrium”. However, the following observations are made:

  1. A lot of grammatical mistakes are found. The authors are suggested to proofread this manuscript.
  2. Page 3 and Table 1 is directly copied from the paper https://www.mdpi.com/2079-9292/10/23/2977/htm
  3. Equation (8) is confusing. It is different from explanation. Can you clarify it?
  4. The previous studies are weak. Compare the previous studies in a table.
  5. Results should compare with the previous studies.
  6. Some acronyms are not defined properly, such as MADDPG, DQN.

Reviewer 2 Report

I would like to congratulate the authors on an interesting paper and research. However, some improvements are still possible.

  1. Not much is written in the paper about the proposed learning algorithm stability and convergence. What criteria did you use to select the parameters of the gradient descent algorithm?
  2. Line 285: Table 1. The pseudo-code of the MPT-NCE method. In the pseudo-code it is not clear whether there are two for loops or only one.
  3. Please write more about the implementation of the algorithms you used in your experiments. Which programming environment did you use? Did you use any special libraries?
  4. Your research would become more important, if the results of your proposed method were compared with the methods in the papers [13]-[17], that you have cited in the References.
  5. In conclusion, can you describe the practical value of presented results and the direction of your further work.

Reviewer 3 Report

I reviewed the paper entitled "A Multi-agent Motion Prediction and Tracking method based on non-cooperative equilibrium." The contribution has merit and is acceptably presented because:

(a) The paper is correctly directed to the proposed target audience of Mathematics, and the organisation of the manuscript flows well.

(b) The references used in this article are appropriate and up to date.

(c) The paper introduces the problem in detail, and the main challenges about self-adaptation in unfamiliar environments are well exposed. The authors provide an interpretation of the results following the domain of the case study. Lastly, they discuss how this outcome is valuable and meaningful.

However, the authors should strengthen the paper before publication. In this regard, I have some remarks to the authors:

  1. Section 1 mentions a series of related studies. Please, add a comparative table. This table would emphasise the novel features of your research.
  2. The title of Section 2 is not descriptive. Perhaps, "Multi-Agent Deep Deterministic Policy Gradient" would fit better.
  3. There is an equation in Line 127; please add its description.
  4. Some equations seem to be wrong (check Eqs. 6, 8, 10). These equations follow the pattern "Something = Something + OtherThing"; however, they imply "OtherThing=0". Please, make a difference between the updated values and the values before updating.
  5. In Eq. 7, what is the difference between Q' (St+1, At+1, θQ') and Q' (st+1, at+1, θQ')?
  6. Citations of studies with several authors are not well written (I mean those citations with "et al."). Check and revise throughout the document.
  7. The paper repeatedly uses the term" mutation". Then, I thought of an evolutionary approach (e.g., a genetic algorithm). However, this assumption seems wrong because the paper does not mention any evolutionary approach. Would you mind clarifying this point?
  8. In Eqs. 11 and 12, the subscripts with three levels (dnd and ini) are unclear. Please, clarify.
  9. In Lines 300–304, the paper reads: "The reward of the detection agents increases when the distance between the detection agents and the interference agents becomes larger. Similarly, the reward of the detection agents decreases when the distance between the detection agents and the interference agents becomes smaller." It is contradictory. Please, correct.
  10. In Lines 353–354, the paper reads: "the prediction rate of the method turned out to be more than 90% based on 100 times repeated prediction experiments." Accordingly, I agree that the proposed MPT-NCE seems better than MADDPG; however, you should conduct non-parametric tests for statistical significance to support this kind of conclusion.
  11. In Section 5, the paper should discuss some directions for future research.
  12. The internal, external, and construct threats were not discussed. You should discuss them in the conclusion section. If you have some complications, then, alternatively, you should at least discuss the scope and the limitations of this research.
  13. It would be best to include (or make publicly available) the code. The purpose of this suggestion is transparency and reproducibility (in fact, the trend in research is to include code and data). Besides, this action may increase the impact of this study in terms of both applications and citations.
  14. You should proofread the document because of some writing mistakes. Additionally, the linguistic quality needs improvement. Ensure that the manuscript reads smoothly; this is essential to help the reader fully appreciate your research. If necessary, consult a professional (e.g., mdpi.com/authors/English, proof-reading-service.com).

Round 2

Reviewer 1 Report

The authors have revised this manuscript accordingly.